# Influence of Multi-Frequency Ultrasound Treatment on Conformational Characteristics of Beef Myofibrillar Proteins with Different Degrees of Doneness

**DOI:** 10.3390/foods12152926

**Published:** 2023-08-01

**Authors:** Zhaoli Zhang, Tingxuan Yang, Yang Wang, Jiarui Liu, Wangbin Shi, Haochen Hu, Yang Meng, Xiangren Meng, Ronghai He

**Affiliations:** 1Key Laboratory of Chinese Cuisine Intangible Cultural Heritage Technology Inheritance, College of Tourism and Culinary Science, Ministry of Culture and Tourism, Yangzhou University, Yangzhou 225127, China; zhaoli@yzu.edu.cn (Z.Z.); 18020226408@163.com (T.Y.); liujiarui_999@163.com (J.L.); sw893934273@163.com (W.S.); 212402505@stu.yzu.edu.cn (H.H.); m2750754998@163.com (Y.M.); 2College of Food Science and Engineering, Yangzhou University, Yangzhou 225127, China; chinawangy@yzu.edu.cn; 3Cuisine Science Key Laboratory of Sichuan Province, Sichuan Tourism University, Chengdu 610100, China; 4School of Food and Biological Engineering, Jiangsu University, Zhenjiang 212013, China; heronghai1971@126.com

**Keywords:** multi-frequency sonication, beef myofibrillar proteins, degree of doneness, structural characteristics

## Abstract

This study evaluated the effect of multi-frequency sonication (20 kHz, 25 kHz, 28 kHz, 40 kHz, 50 kHz) on structural characteristics of beef myofibrillar proteins (MPs) with different degrees of doneness (Rare 52~55 °C, Medium Rare 55~60 °C, Medium 60~65 °C, Medium Well 65~69 °C, Well Down 70~80 °C, and Overcooked 90 °C). The results showed that surface hydrophobicity and sulfhydryl content increased with the increase in degree of doneness. At the same degree of doneness, the sulfhydryl group contents reached the maximum at a frequency of 28 kHz. In addition, the absolute value of ζ-potential was significantly decreased after ultrasonic treatment (*p* < 0.05). SDS gel electrophoresis showed that the bands of beef MPs were not significantly affected by various ultrasonic frequencies, but the bands became thinner when the degree of doneness reached overcooked. Fourier transform infrared spectrum showed that with the increase of ultrasonic frequency, α-helix content decreased, and random coil content significantly increased (*p* < 0.05). The results of atomic force microscopy indicated that the surface structure of beef MPs was damaged, and the roughness decreased by sonication, while the roughness significantly increased when the degree of doneness changed from medium to overripe (*p* < 0.05). In conclusion, multi-ultrasound combined with degree of doneness treatment alters the structural characteristics of beef MPs.

## 1. Introduction

In recent years, China’s economy has been steadily developing, and people’s living standards have been improving; beef has entered the general public’s kitchen. As a kind of high-quality meat food with more lean meat and less fat, it is gradually becoming more valued and favored by people, and more and more beef products are also welcomed by the market. Beef has many cooking methods, among which fried steak is favored by beef lovers. To ensure the unique tenderness of steaks, Western food especially divides the maturity of steaks into the following six degrees: Rare (52~55 °C), Medium Rare (55~60 °C), Medium (60~65 °C), Medium Well (65~69 °C), Well Down (70~80 °C) and Overcooked (90 °C) [1].

There is a high protein content in beef, with myofibrillar proteins (MPs) accounting for 50–60% of the total protein in muscle, and the protein content can exceed 70% for beef [2]. It is an indispensable main component in the production process of emulsified minced meat products. MPs are a kind of mixed protein, mainly composed of actin, myosin, regulatory protein, and actin [3]. Due to their unique, essential amino acid content and superior biological digestibility, MPs have been extensively studied [4,5,6]. Aside from that, MPs have also been considered individual building blocks to help construct gels and delivery systems that are suitable for modern consumers in terms of health benefits and quality of food [7,8]. Due to their flexible and sensitive structure, meat proteins have gained a great deal of attention as functional food ingredients.

MPs under the condition of hot or cold will have obvious changes, such as MPs under heating conditions can form a gel network structure related to the crosslinking reaction happened, and under the condition of frozen storage, MPs degradation reaction happens itself; hot and cold conditions will affect the stability of the myofibril protein [9]. Therefore, the scientific research of myofibrillar protein in beef is very important and can provide corresponding guiding significance for the development of the beef processing industry.

Myofibrillar proteins can be modified in several ways by green processing techniques today. According to Cai et al. [10], microwave treatment could alter the conformational characteristics of fish myofibrillar proteins. Wang et al. [11] indicated that pulsed electric field treatment helped to alter the tenderness and biophysical attributes of pork myofibrillar protein. Also, Liu et al. [12] observed that ultra-high-pressure treatment greatly affected the oxidation and gel characteristics of fish myofibrillar proteins. Meanwhile, ultrasonic technology, as a green processing technology, has been widely utilized in the food industry. There was widespread use of ultrasonic technology in meat processing that was non-thermal physical processing technology. Ultrasound has gained more attention ascribed to its distinctive cavitation effect and high efficiency [13,14,15]. Specifically, the low-frequency and high-intensity ultrasound with the frequency of 20~50 kHz and the intensity of more than 1 W·cm^−2^ could effectively enhance the quality of the meat and the processing efficiency, which has a positive role in promoting the development of the meat processing industry. It modified macromolecules in food by mechanical cavitation heat and chemical effects, which could not only change their physical and chemical properties but also alter their structure, with remarkable effects [16,17]. Thus, this technology has become a focus for research and development, with huge potential [18,19]. Based on Cheng et al. [20], dual-frequency ultrasound treatment had a more significant effect on the structure of whey protein, leading to better physicochemical characteristics. Amiri et al. [21] explored that ultrasound treatment could successfully improve the functional and physicochemical properties of MPs and affect their rheological properties. Li et al. [22] showed that the emulsifying activity index and emulsifying stability index of chicken myofibrillar proteins were significantly increased after ultrasonication, and a more stable emulsion could be formed from the myofibrillar proteins. However, there is little information on the impact of different ultrasound frequency combinations on the structural characteristics of the protein(s).

Therefore, this article aims to explore the effects of five-frequency ultrasound operating modes on the aggregation and structural characteristics of MPs with different degrees of doneness. The protein structural characterizations were assessed by surface hydrophobicity, sulfhydryl group, SDS-PAGE gel electrophoresis, particle size, and ζ-potential, Fourier transform infrared spectrum (FITR), and Atomic Force Microscope (AFM). This work provides guidance on the development of ultrasonic technology applicable to the production of meat products for the food industry.

## 2. Materials and Methods

### 2.1. Materials 

Xinjiang brown cattle premium organic eye steak was purchased from METRO Supermarket (Xinjiang Tianlai Breeding Co., Ltd., Yangzhou, China). The excess fat and connective tissue should be removed and stored in a low-temperature refrigerator at −20 °C until needed. The meat samples were thawed at 0–4 °C for 2 h before extraction of myofibrillar protein. Experiments were conducted with analytical-grade reagents.

### 2.2. Extraction of Beef MPs

The extraction of MPs refers to the method of Doerscher et al. [23] with modifications. The fresh beef steak was provided by METRO Supermarket (Yangzhou, China). MPs were extracted from the beef steak at a low temperature (0–4 °C) by removing visible connective tissue. The beef steak (5 g) was homogenized with 4-time volumes of a solution (0.1 mol/L KCl, 20 mmol/L Tris-HCl, pH = 7.0) using a homogenizer (Ultra Turrax T25 BASIS, IKA Company, Hamelin, Germany) at a speed of 10,000 r/min in an ice bath for 60 s. The homogenate was centrifuged at 2000× *g* for 10 min at 4 °C. It was suspended in four times volumes of A solution, homogenized, and centrifuged twice. The precipitate was homogenized in an ice bath for 60 s at a speed of 10,000 r/min with 3 times the volume of B solution. Using 2000× *g* at 4 °C for 15 min, the homogenate was centrifuged. A solution was homogenized with four times the precipitate volume and centrifuged once. Extracts were stored at −20 °C for 48 h.

### 2.3. Ultrasound Treatment and Insulation Treatment of Beef MPs

MPs were placed in the chamber of Ultrasonic equipment (Figure 1, RC-1000LG, Renchuan Company, Langfang, China) and pretreated using various sonication frequencies. The experiment was conducted under the following conditions: ultrasound power density of 100 W/L; ultrasound time of 5 min; ultrasound temperature of 30 °C; and substrate concentration was 50 g/L. Subsequently, MPs were added 10 mL to the test tube at room temperature for 2 h; then, samples were placed in constant temperature water bath at 54 °C (Rare), 58 °C (Medium Rare), 63 °C (Medium), 68 °C (Medium Well), 72 °C (Well Done), and 100 °C (Overcooked) for 10 min immediately. Control (MPs sample) was prepared with the same conditions but without ultrasonication. After ultrasonication, all of the samples were stored at 4 °C for further observation.

### 2.4. Determination of Surface Hydrophobicity (SUH)

In order to determine the surface hydrophobicity (SUH) of MPs, we used a multifunctional enzyme labeling instrument (INFINITE 200 PRO Tecan Company CH, Mannedorf, Switzerland). In order to facilitate better dispersion, different protein samples were diluted with buffer to about 1 mg/mL. After heat treatment, it was determined by referring to the method of Chelh et al. [24]. After that, 1 mg per mL of bromophenol blue solution (40 μL) and the suspension (1 mL) were mixed and left to react in the dark (25 °C) for 10 min. For the blank control, 1 mg/mL bromophenol blue solution (40 μL) and the extraction buffer (1 mL) were mixed and reacted in the dark for 10 min (25 °C). The substance was centrifuged by 4000 g for 15 min after the reaction (4 °C), and measure the absorbance at 595 nm. The hydrophobicity formula is as follows:SUH = 40 × (OD_control_ − OD_sample_)/OD_control_(1)
where SUH is the surface hydrophobicity (μg); OD_control_ is the absorbance value of the blank; OD_sample_ is the absorbance value of the sample.

### 2.5. Determination of Sulfhydryl Group 

Multifunctional enzyme labeling equipment (INFINITE 200 PRO Tecan Compay, Mannedorf, Switzerland) was used to measure the sulfhydryl group of MPs. In order to determine the total sulfhydryl content and free sulfhydryl content of all MPs, absorbance measurements were conducted (412 nm). Ellman’s reagent (100 μL, DTNB in buffer) and MP solution (4 mL, 1 mg/mL) were mixed in the dark. The free sulfhydryl contents and total sulfhydryl contents of the MP samples were defined as follows: (2)$$\mathrm{ASC}\left(\mu \mathrm{mol}/100\;\mathrm{mg}\right)=\frac{A\times C}{E_{M}}$$
where A is the absorbance of MPs, C is the protein concentration, and $E_{M}$ is the molar extinction coefficient (13,600).

### 2.6. SDS-PAGE Gel Electrophoresis

MP solution was diluted to 1 mg/mL using 20 mmol/L Tris-HCl buffer (containing 0.6 mol/L KCl) prior to gel running [25]. Moreover, 12% acrylamide gels were used for separating, and 5% acrylamide gels were used for stacking. Electrophoresis was carried out in a vertical (MiniPROTEAN^®^ Tetra System, BIO-RAD, Hercules, CA, USA) unit with buffer (50 mmol/L Tris-HCl, 384 mmol/L glycine, and 0.1% SDS, pH = 8.3). At a stacking voltage of 80 mV and a separation voltage of 120 mV, ten microliters of the results were injected into stacking gel. Protein bands were fixed and then stained by Coomassie Brilliant Blue (R-250, BIO-RAD, CA, USA) for 1.5 h. A solution containing 5% methanol (*v*/*v*) and 7.5% acetic acid (*v*/*v*) was used for decoloration.

### 2.7. Fourier Transform Infrared Spectra (FTIR) 

As described by Han et al. [26], KBr was mixed with the sample (1:100–1:200), milled to 2–3 μm, and then compressed into tablets/pellets. The AVATAR FT-IR spectrometer (TENSOR 27, BRUKER Co., Bremen, Germany) was used (wavenumber range 4000–400 cm^−1^, 128 scans). An average of the scans was taken, and background correction was performed using the KBr spectrum. PeakFit software v4.12 was used to analyze the change in protein secondary structure observed by deconvolution of the amide I band (1600–1700 cm^−1^).

### 2.8. Particle Sizes Determination

LitesizerTM 500 nm laser particle size analyzer (Microtrac S3500, Largo, FL, USA) was used to determine the particle size of MPs. Samples were diluted to the concentrations set (1 mg/mL). Then, 2 mL of prepared proteins were added to the sample box. It is possible to determine directly the particle size of MPs and their particle size distribution by using the instrument.

### 2.9. ζ-Potential Determination 

A different instrument was used to evaluate the ζ-potential of prepared MPs (Malvern Instruments, ltd., Great Malvern, UK). Briefly, 0.5 mL of prepared proteins (1 mg/mL) were added to the sample box. It was calculated by analyzing the frequency shifting of the light scattered by charged proteins. Final results were analyzed by the instrument. 

### 2.10. Atomic Force Microscope (AFM)

A multimode microscope (Bruker, Santa Barbara, CA, USA) was used to examine the microstructure and surface roughness. The AFM observations were carried out with samples in PeakForce QNM mode using the Bruker Scan Asyst needle (300 kHz, 25.1 N/m).

### 2.11. Statistical Analysis

All data were performed with three determinations. The Origin software (32-bit, Version 9.65) and the statistical product and service solutions (SPSS) statistics software (version 20.0) were utilized to analyze the data and visualize the correlations. Differences between the samples were compared using one-way analyses of variance (ANOVA) and Duncan’s test (*p* < 0.05).

## 3. Results and Discussions

### 3.1. Surface Hydrophobicity

Surface hydrophobicity (SUH) is an indicator used to evaluate the structural alteration in the protein network. The SUH affected the interaction between protein molecules; thus, bromophenol blue was used to sign the hydrophobic alterations on the surface of MPs. As seen in Figure 2, multi-frequency ultrasound pretreatment was applied to the cooking reaction of MPs with different degrees of doneness. The results showed that ultrasound pretreatment could significantly enhance the surface hydrophobicity (*p* < 0.05) under the conditions of various cooking doneness. In addition, with the increase of ultrasound frequency, SUH firstly increased and then decreased with further treatment (ultrasound frequency 50 kHz). The SUH was the least at the ultrasound frequency of 50 kHz when the degree of doneness was Rare (52–55 °C). In this case, excessive ultrasound frequency may cause cavitation, which re-embeds hydrophobic amino acids left on the protein surface. The conclusion of the study by Bian et al. [27] is similar.

When the ultrasound frequency was 28 kHz, SUH reached the maximum value. This is attributed to the fact that sonication may induce the unfolding and conformational change of proteins, allowing hydrophobic regions to be transferred from MPs’ interior to their surface. Other studies have also reached a similar conclusion. Chen et al. [5] reported that ultrasonic treatment by different treatments (20 kHz, 23 kHz, and 20/23 kHz) induced an increase in the SUH of all MP samples, implying the exposure of hydrophobic amino acids. Deng et al. [28] revealed that compared to those of the control, the surface hydrophobicity (S_o_-ANS) of all high-intensity ultrasonication (HIU, 100, 150, 200, and 250 W)-treated samples were significantly higher. This phenomenon might be due to HIU-promoted MPs molecule unfolding and stretching, and the partially buried hydrophobic groups within were exposed, thus increasing the surface hydrophobicity. The results were consistent with previous studies by Amiri et al. [21] and Wang et al. [29], who investigated similar results for beef myofibrillar proteins and yellow croaker myofibrillar proteins.

What is more, at the same ultrasound frequency, the SUH of treated MPs was enhanced with the increasing degree of doneness and reached its maximum at Medium Well (65~69 °C, Figure 2D). Afterward, with further increases in the degree of doneness, such as Well Down (70~80 °C, Figure 2E) and Overcooked (90 °C, Figure 2F), the SUH value displayed a decrease. The observed results about the improvement in SUH indicated that the degree of doneness could cause the protein molecules unfolding, and destroy the hydrophobic interactions among molecules, leading to the exposure of hydrophobic groups/regions in the protein molecules. However, it was obvious that the degree of doneness might affect the degeneration of myosin, and MPs accumulation and reaction with side chains, then, as a result of precipitation. A small amount of hydrophobic linkages/bonds are re-embedded in myosin due to protein aggregation, leading to a reduction in SUH values [30]. This finding was in similitude to the report of Promeyrat et al. [31]. Moreover, another study by Wang et al. [32] pointed out that most surface hydrophobic groups were located at the head of myosin, and the hydrophobicity of myofibrils increased during heating temperature. Later, relevant studies have also reported that the surface hydrophobicity of proteins from various sources is varied due to their different structures. Overall, sonicated MPs at various degrees of doneness in a wide temperature range may have altered the surface hydrophobicity, and these changes were positively associated with functional utilizations and proteins conformation.

### 3.2. Sulfhydryl Content

Sulfhydryl is generally acknowledged as a crucial indicator to reflect the conformational alternation in the protein molecules. The sulfhydryl group in myofibrils is readily oxidized to disulfide bonds, leading to a reduction in the sulfhydryl content. Thus, the sulfhydryl content can evaluate the extent of protein oxidation [12]. The sulfhydryl content of sonicated and non-sonicated MPs via different degrees of doneness is given in Figure 3. Figure 3 displays that the sulfhydryl content at various treatments significantly increases following ultrasonication (*p* < 0.05), consistent with the literature of Zhao et al. [33], and this trend was not changed by the alteration in the degree of doneness. Notably, the total sulfhydryl content has the maximum value at a frequency of 28 kHz, compared to the control, and it increased by 23.53%, 27.78%, 23.58%, 11.78%, and 24.36%, respectively, at various degrees of doneness (Rare, Medium Rare, Medium, Medium Well, Well Done, Figure 3A–E). Additionally, the free sulfhydryl content showed a similar observation in this study. MPs are rich in thiol groups, which are readily converted into intermolecular and intramolecular disulfide bonds under the influence of some factors, such as ultrasound treatment. Also, ultrasonic cavitation may cause the MP structure to unfold to generate more active groups, leading to exposure of the reactive sulfhydryl group to the protein surface, resulting in the enhancement of sulfhydryl content. Furthermore, the improvement of free sulfhydryl content might be owing to the high pressure and shear force caused by ultrasonication, leading to the destruction of disulfide bonds in protein molecules and the generation of more free sulfhydryl group. A similar finding was also reported by Feng et al. [27].

Following sonication, with the continuous increase of the degree of doneness, the total sulfhydryl group and the free sulfhydryl group of MPs first increased and then decreased and reached the maximum value at the Medium Well (Figure 3D). Studies have indicated that heating temperature significantly affects the degree of actomyosin dissociation, and lower heating temperature (50–60 °C) can promote the dissociation of actomyosin to different degrees [34,35]. In this study, it was found that the content of the active thiol group of myofibrils increased from the control group to the Medium Rare, indicating the dissociation of actomyosin molecules. This may be because the terminal temperature of the protein was about 58 °C at the Medium Rare, which was just in the optimal temperature range for actomyosin dissociation. Nevertheless, as the degree of doneness was increased, the actomyosin conformation was altered, and the active sulfhydryl group was oxidized to disulfide bonds, leading to a decrease in sulfhydryl content. 

### 3.3. SDS-PAGE 

Figure 4 displays the effects of multi-frequency ultrasound treatment on the primary structure of MPs with various degrees of doneness. From the SDS-PAGE gel electrophoresis (Figure 4), it can be seen that there are seven distinct bands on the MPs swimming lanes treated by varying degrees of doneness, from top to bottom, mainly including myosin heavy chain, α-actin, myosin light chains 1, tropomyosin, myosin I subunit, myosin light chains 2, and myosin light chains 3. The bands and mobility of the electrophoretic spectrum of MPs treated by sonication were similar to these of non-sonicated MPs. Also, this may be because, with the increase in ultrasound frequency, the distribution of protein composition and molecular weight was not altered significantly following ultrasonication. The observed phenomenon of primary structure may be due to the cavitation and catalytic effects caused by ultrasound [36]. During ultrasound, bubbles are generated in liquid foods, and the bubble size and implosion strength are determined by the frequency [37]. However, although sonication at the low frequency used in this study (20 kHz, 25 kHz, 28 kHz, 40 kHz, 50 kHz) causes micro-streaming currents and cavitation [37,38], the compression caused by sound waves is insufficient to separate the liquid molecules. Overall, sonication at lower frequency did not alter protein composition and the subunits of MPs.

In addition, as shown in Figure 4, myosin heavy chain (>180 kDa) and actin (130 kDa) were known as the major bands, which also verifies the view that myosin and actin are the main components of myofibrillar protein [39]. Generally, the degradation of protein molecules is manifested as blurring, weakening, and expansion of bands at higher molecular weights, as well as new bands or darkening bands. This finding was in similitude to the work of Wang et al. [38]. Their study displayed that ultrasonication treatment induced no noticeable alteration in rice dreg protein subunits (20 kHz, 28 kHz, 35 kHz, 40 kHz, and 52 kHz). Likewise, Zhu et al. [40] reported that walnut protein isolates did not alter in molecular weight following ultrasound treatment (200 W, 15–30 min) followed by SDS-PAGE analysis. 

Furthermore, as the frequency of sonication was unified, the results of this study suggested that the MPs experienced a few alterations at molecular weights with various degrees of doneness. Compared to the control, myosin heavy chain and actin degradation were evident, as reflected in the increase firstly and then the decrease in intensity of the bands. As well, with the increasing intensity of the bands, myofibrillar proteins aggregated, resulting in an increase in their molecular weight distribution. Later, MPs were treated by continuing to heat until they reached their overripe state; the protein molecules were completely denatured, leading to protein degradation. So, the molecular weights of MPs decreased.

### 3.4. Secondary Structure

During nascent protein maturation in an aqueous solution, the primary polypeptides coiled distinctively, forming a secondary structure. The secondary structure of protein can be characterized by the ratio of the α-helix, β-sheet, β-turn, random coil, and unordered groups. The α-helix is the ordered arrangement of protein molecules maintained by intramolecular hydrogen bonds, and the β-sheet is the ordered arrangement of protein compartments maintained by intermolecular hydrogen bonds. The α-helix, β-sheet, β-turn, and random coil contents of MPs under various ultrasonic frequencies are displayed in Figure 5A. As ultrasonic frequency increased, there was a decrease in α-helix content and a significant increase in random coil content (*p* < 0.05). Upon 40 kHz ultrasonic treatment, the proportion of β-sheet and β-turn content showed distinct alterations, reaching a maximum (50%), and a minimum (34%), respectively. The results indicated that ultrasonic treatment induced a certain degree of molecular unfolding of the proteins. Furthermore, as the degree of doneness reached Well Done (UT6, 72 °C), there was no significant difference in the secondary structure of MPs (*p* > 0.05). Stathopulos et al. [41] reported that ultrasonication-induced protein aggregation exhibited a high level of β-structure content, and proteins with substantial amounts of native α-helix structures displayed an increase in β-structure after ultrasonication and a concomitant decrease in α-helix structure in the aggregation. Moreover, the study of Lee et al. [42] pointed out that the participation of β-sheet in the secondary structure played a role in protein aggregation and network formation.

Figure 5B−H displayed the FTIR spectra of beef MPs at ultrasound combined with various degrees of doneness. Significant peaks were observed within the range of 1700−1600 cm ^−1^ (amide I band), including the stretching vibration of C=O, stretching vibration of C−N, bending vibration of C_α_−C−N, and in-plane bending vibration of N−H. Generally, the bands at 1646−1664 cm^−1^, 1664−1681 cm^−1^, 1682−1700 cm^−1^, and 1637−1645 cm^−1^ correspond to α−helix, β−turn, β−sheet, and random coil, respectively [43]. 

As shown in Figure 5B–H, the secondary structure of beef MPs treated with different treatment methods was significantly varied (*p* < 0.05). Compared with the control group, the significant peaks of each sample group after ultrasonic treatment were shifted to the right. The results indicated that cavitation caused by ultrasonic treatment reduced the degree of aggregation and increased the stability of MPs. In addition, with the increase in ultrasonic frequency, the significant peaks of each treatment group were shifted to the right, and the intensity increased first and then decreased. When the ultrasonic frequency was 25 kHz, the characteristic absorption peak intensity reached the maximum.

As can be seen from Figure 5B–H, with the increase in the degree of doneness, the intensity of significant peaks of UT6 samples was significantly lower than in other treatment groups (*p* < 0.05). The results indicated that heat treatment would denature proteins and degrade some proteins. This is consistent with the results of Zheng et al. [44].

### 3.5. Effect of Ultrasound Treatment on the Aggregation Behavior of MPs with Different Degrees of Doneness

#### 3.5.1. Analysis of Particle Size 

The particle size measurement revealed protein aggregation, which played a distinguished role in the functional characteristics [15]. The alteration of particle size was mainly induced by the cross-linking and aggregation of protein molecules [45]. As shown in Figure 6A, after ultrasonic treatment, the particle size of MPs generally shifted significantly to the right (*p* < 0.05). The results may be attributed to the interruption of the non-covalent bonds between MPs by the physical effects of sonication cavitation and microbeam effect [46,47]. As ultrasonic frequency increased, the particle size of MPs first increased and then decreased, reaching the maximum size at 40 kHz, indicating that MPs were more beneficial for protein accumulation. Furthermore, the particle size distribution of the UT6 group overall shifted to the right, and the peak distribution width of the UT6 group was narrower than that of the UT5 group. This phenomenon indicated that high-temperature treatment degraded MPs aggregates and reduced the average particle size. This result is in similitude to the outcomes of Zheng et al. [44].

#### 3.5.2. Analysis of ζ−Potential

ζ−potential indicates the surface charge of suspended particles, which is an important parameter for protein stabilization [48]. Higher absolute ζ−potential implies that the solution is difficult to aggregate, whereas lower absolute ζ−potential makes aggregation easier [49]. As shown in Figure 6B, sonication−assisted−various degree of doneness treatment has a significant influence on the ζ−potential of beef MPs (*p* < 0.05). The ζ−potential of MPs samples was negative, indicating that the negative amino acids outnumbered the positive amino acids on the protein surface. Zhang et al. [50] showed that the proteins with higher ζ−potential (negative or positive) were electrically stable, while proteins with lower ζ−potential were more tend to aggregate. 

Compared with the control group, the absolute value of ζ−potential decreased significantly after ultrasonic treatment. This may be due to the aggregation effect of the cavitation induced by ultrasonic treatment on beef MPs, resulting in the exposure of protein molecules and reduced surface negative charge, which is consistent with the research results of Zhang et al. [51]. In addition, beef MPs treated in the UT6 group displayed the largest absolute potential, indicating that higher temperatures would destroy the aggregation of MPs and increase the exposure of charged groups on the surface of MPs, consistent with the conclusion by Zheng et al. [44].

### 3.6. Effects of Ultrasound Treatment on the Microstructure of MPs with Different Degrees of Doneness

Atomic force microscopy (AFM) is considered a useful technique to exhibit the microscopic morphology of proteins at the nanometer level, which is always closely associated with their functional characteristics. In general, the microcosmic morphology of free actin is closely related to actomyosin dissociation, and the microstructure images and height distribution of free actin can be obtained by using AFM [52]. 

The microstructure of beef MPs in different treatment groups is shown in Figure 7. Compared with the control, the roughness of beef MPs after ultrasonic treatment was significantly reduced. This is similar to the study by Fu et al. [53]. This phenomenon may be attributed to the strong physical forces produced by ultrasonication cavitation, including shear force, shock wave, and turbulence. Protein particles can be effectively broken down and minified in size by these physical forces c [49].

When the ultrasonic frequency was increased to 28 kHz, the particle distribution of beef MPs was uniform, and the roughness was significantly reduced, which indicated that the formation of protein aggregates could be most inhibited when the ultrasonic frequency was 28 kHz. In addition, compared with UT5 (50 kHz, 64 °C), the particle number and uneven particle distribution roughness of UT6 (50 kHz, 72 °C) increased significantly, which may be because the high temperature promoted the denaturation of beef myofibrillar protein and the formation of protein aggregates.

## 4. Conclusions

In this study, beef MPs were modified via multi-frequencies ultrasound (20 kHz, 25 kHz, 28 kHz, 40 kHz, and 50 kHz) with various degrees of doneness. The results showed that multi-frequencies ultrasound combined with the degree of doneness treatment significantly affected the aggregation behavior and structural characteristics of beef MPs. When the ultrasonic frequency was 28 kHz, the surface hydrophobicity of MPs under the degree of doneness (Medium Well) was the largest. In addition, sulfhydryl group contents reached the largest value under r the degree of doneness (Rare, Medium Rare, Medium). The surface hydrophobicity and sulfhydryl content also improved with the increase in the degree of doneness. The structural properties results analysis indicated that ultrasonic treatment could decrease the absolute value of ζ−potential and the surface roughness and damage the surface structure of beef MPs with various degrees of doneness. In conclusion, appropriate ultrasonic frequency and degree of doneness can alter the structural characteristics of beef MPs. Further research is needed to analyze the changes in nutrient composition and effects on digestion and absorption characteristics in proteins undergoing sonication combined with the degree of doneness treatment.

## Figures and Tables

**Figure 1 foods-12-02926-f001:**
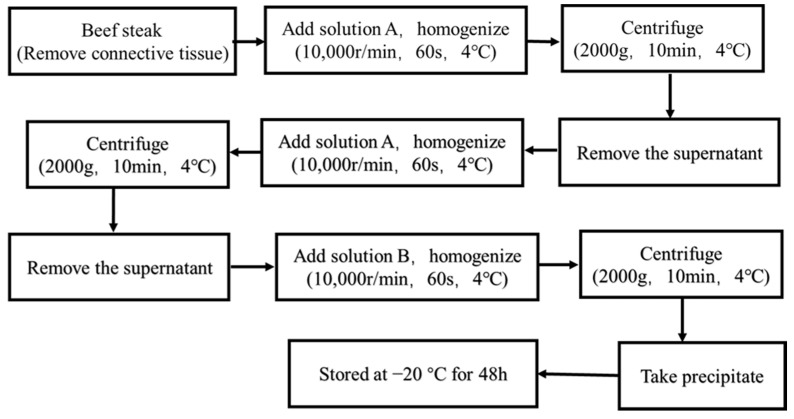
The schematic diagram of the ultrasound device.

**Figure 2 foods-12-02926-f002:**
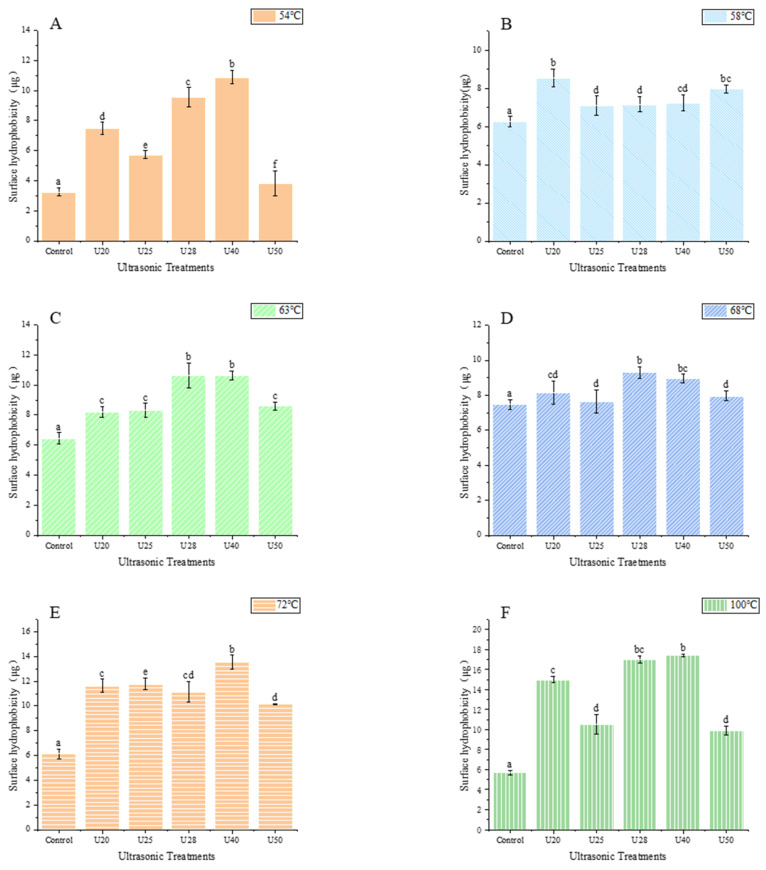
Effects of Surface hydrophobicity of myofibrillar protein at different ultrasonic frequencies Different small letters represent significant differences(*p* < 0.05). (**A**) 54 °C; (**B**) 58 °C; (**C**) 64 °C; (**D**) 68 °C; (**E**) 72 °C; (**F**)100 °C.

**Figure 3 foods-12-02926-f003:**
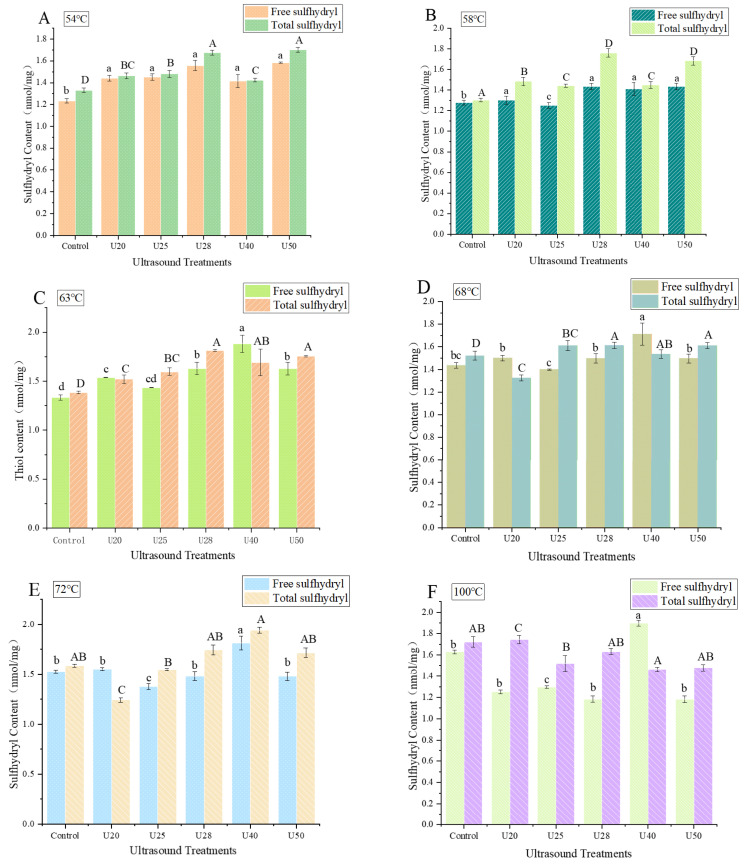
Sulfhydryl content of myofibrillar protein at different ultrasonic frequencies. U20: ultrasound frequency of 20 kHz; U25: ultrasound frequency of 25 kHz; U28: ultrasound frequency of 28 kHz; U40: ultrasound frequency of 40 kHz; U50: ultrasound frequency of 50 kHz. Different capital or small letters represent significant differences(*p* < 0.05). (**A**) 54 °C; (**B**) 58 °C; (**C**) 64 °C; (**D**) 68 °C; (**E**) 72 °C; (**F**) 100 °C.

**Figure 4 foods-12-02926-f004:**
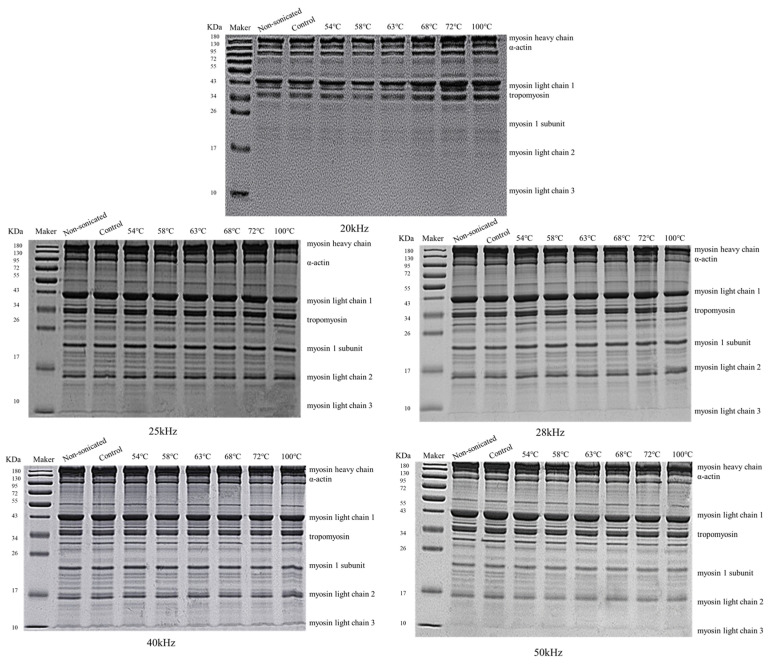
SDS-PAGE of MPs treated by ultrasound frequency of 20 kHz, 25 kHz, 28 kHz, 40 kHz, and 50 kHz.

**Figure 5 foods-12-02926-f005:**
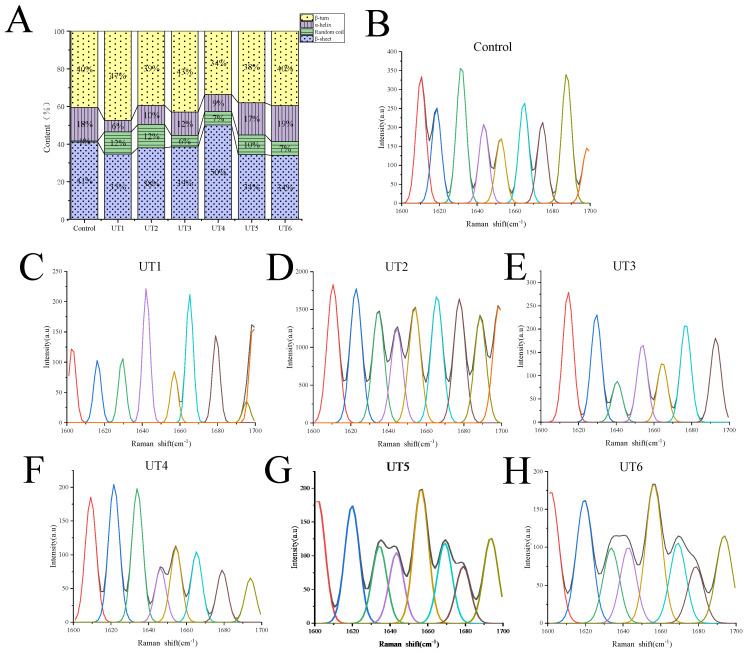
Secondary structure of MPs under different treatments (**A**), Raman spectra of MPs treated by different ultrasound frequencies treatments. (**B**) Control; (**C**) UT1 (ultrasound frequency of 20 kHz, 64 °C); (**D**) UT2 (ultrasound frequency of 25 kHz, 64 °C); (**E**) UT3 (ultrasound frequency of 28 kHz, 64 °C); (**F**) UT4 (ultrasound frequency of 40 kHz, 64 °C); (**G**) UT5 (ultrasound frequency of 50 kHz, 64 °C); (**H**) UT6 (ultrasound frequency of 50 kHz, 72 °C).

**Figure 6 foods-12-02926-f006:**
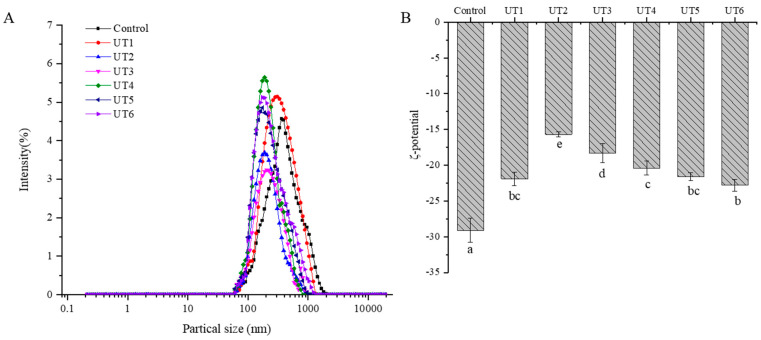
Effect of ultrasonic frequency on ζ−potential (**A**) and particle size (**B**) of myofibrillar protein at different degrees of doneness. UT1: ultrasound frequency of 20 kHz, 64 °C; UT2: ultrasound frequency of 25 kHz, 64 °C; UT3: ultrasound frequency of 28 kHz, 64 °C; UT4: ultrasound frequency of 40 kHz, 64 °C; UT5: ultrasound frequency of 50 kHz, 64 °C; UT6: ultrasound frequency of 50 kHz, 72 °C. Different small letters represent significant differences(*p* < 0.05).

**Figure 7 foods-12-02926-f007:**
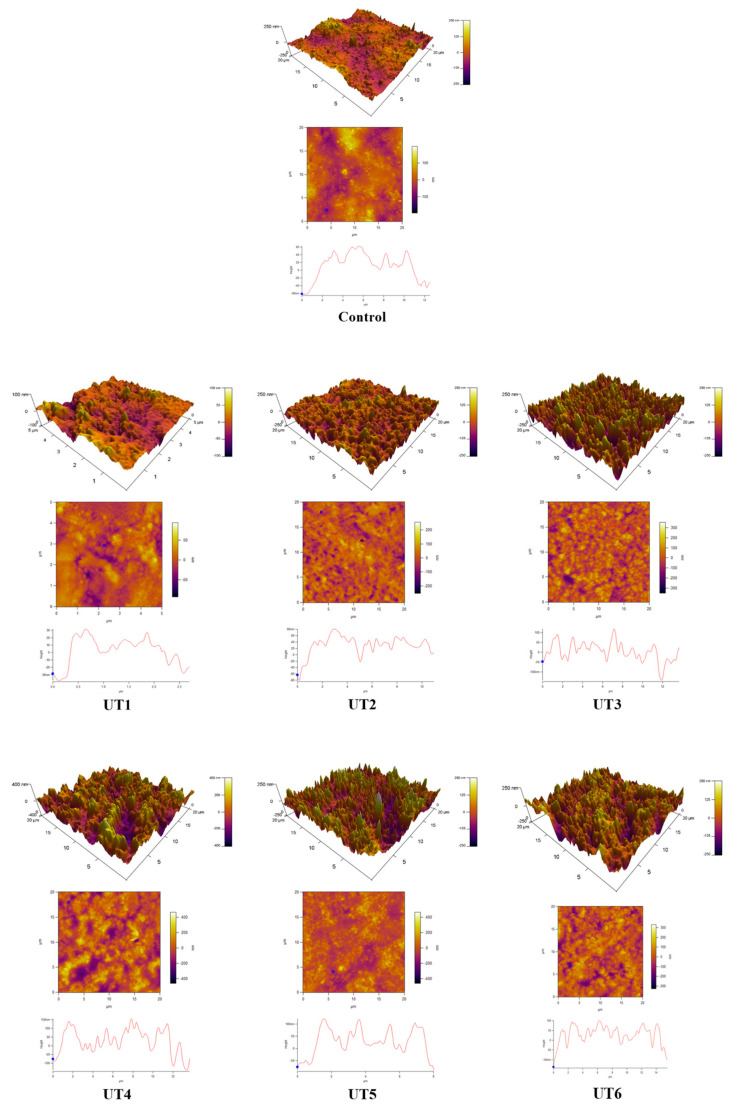
Effect of ultrasonic frequency on surface morphology of beef myofibrillar proteins at different degrees of doneness. UT1: ultrasound frequency of 20 kHz, 64 °C; UT2: ultrasound frequency of 25 kHz, 64 °C; UT3: ultrasound frequency of 28 kHz, 64 °C; UT4: ultrasound frequency of 40 kHz, 64 °C; UT5: ultrasound frequency of 50 kHz, 64 °C; UT6: ultrasound frequency of 50 kHz, 72 °C.

## Data Availability

Data is contained within the article.

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
