# Peer review of "Influence of Multi-Frequency Ultrasound Treatment on Conformational Characteristics of Beef Myofibrillar Proteins with Different Degrees of Doneness"

_foods, 2023, doi:10.3390/foods12152926_

Round 1

Reviewer 1 Report

This is an interesting paper that assesses Influence of Multi-Frequency Ultrasound Treatment on Conformational Characteristics of Beef Myofibrillar Proteins with Different Degree of doneness. It is well organized and the results are interesting. While I have few comments for improving readability and emphasizing key results, the manuscript is worth consideration after addressing these comments and revising the manuscript.

1.     As feedstock for the experiments, Multi-Frequency Ultrasound Treatment was used to Influence on Conformational Characteristics of Beef Myofibrillar Proteins. One of my initial concerns was whether it would be possible to accurately identify and pinpoint a necessity of multi frequency treatment as well as a base for a chosen frequencies 20 kHz, 25 kHz, 28 kHz, 40 20 kHz, 50 kHz. Why this frequency range? Any fundamental background? This will make the motivation for this experiment easier to understand in my opinion and introduce the reader better into the topic.

2.     Something else missing from the Introduction is a correlation with papers based on other green technologies: high hydrostatic pressure, Pulsed electric field etc.

3.     M&M overall. Due to the high amount of processes I suggest to add an illustrative scheme of Process layout for the MPs. This will make easier to understand in my opinion M&M section.

4.     Figure 1. The schematic diagram of the ultrasound device: I did not understand where is a US nozzle? Please indicate which part of your device was connected to US nozzle. Additionally please clarify which part of your chamber was connected to US nozzle (walls or bottom?).

5.     Figure 2. Please add more discussion regarding U50 (T=54 °C). Why does Surface hydrophobicity drops and this frequency level? Any oscillations?

6.     Fig 3 and 5. There are several samples indicated as: UT1…UT6 and U20….U50 please clarify.

7.     Line 416: The authors stated: When the ultrasonic frequency was 28 kHz, the surface hydrophobicity and sulfhydryl group contents of MPs under the same degree of doneness were the largest. However fig 2 do not state this information. Please clarify.

Based on the above points, I would propose a major revision of the manuscript.

Author Response

Response to reviewer #1’s comments

Comments and Suggestions for Authors

This is an interesting paper that assesses Influence of Multi-Frequency Ultrasound Treatment on Conformational Characteristics of Beef Myofibrillar Proteins with Different Degree of doneness. It is well organized and the results are interesting. While I have few comments for improving readability and emphasizing key results, the manuscript is worth consideration after addressing these comments and revising the manuscript.                                                                                   

Q1. As feedstock for the experiments, Multi-Frequency Ultrasound Treatment was used to Influence on Conformational Characteristics of Beef Myofibrillar Proteins. One of my initial concerns was whether it would be possible to accurately identify and pinpoint a necessity of multi frequency treatment as well as a base for a chosen frequencies 20 kHz, 25 kHz, 28 kHz, 40 kHz, 50 kHz. Why this frequency range? Any fundamental background? This will make the motivation for this experiment easier to understand in my opinion and introduce the reader better into the topic.

Answer: Thanks for your review and kind advice.

The ultrasonic process is a non-thermal physical processing technology based on mechanical vibration of the medium that produces a frequency of more than 20 kHz. Through the transducer, ultrasonic waves convert acoustic energy into mechanical energy, thereby causing the elastic medium to vibrate and cause physical and chemical effects. As a result of the ultrasonic effect, thermal, cavitation, and mechanical effects are primarily responsible, and the frequency and intensity of the ultrasonic effect determine the application range. According to the frequency and intensity and in practical application, ultrasound can be divided into two categories, one is high-intensity low-frequency ultrasound (power intensity > 1 W/cm2, frequency 20 kHz-100 kHz), which has high energy, can make the medium inside the production of fastmoving microbubble flow, after a series of changes in the final bubble violent rupture, Thus affecting the product processing process. The other is low-intensity high-frequency ultrasound (power intensity < 1 W/cm2, frequency 100 kHz -1 MHz), which contains low energy, usually used for food composition characterization, food structure improvement, etc., this measurement process does not alter physical or chemical properties of food materials, nor does it cause any loss of the product.

The low frequency ultrasound(20-50 kHz)can be widely applied in food processing, such as freezing, drying, defoaming, enzyme activation and inactivation, etc. In solid-liquid systems of ultrasound pretreatment prior to drying, the ultrasound wave can induce a series of comprehensive effects, including acoustical cavitation, sponge effect, acoustic streaming, and microscopic channels, which greatly affect the mass transfer and cell wall permeability. Moreover, low frequency ultrasound (20-50 kHz), which is generally used as protein modification treatment, with the advantages of easy operation, high efficiency, and applicable to large-scale applications. As a result, ultrasound frequencies were designed for 20 kHz, 25 kHz, 28 kHz, 40 kHz, and 50 kHz in the ultrasound equipment for this study.

Q2. Something else missing from the Introduction is a correlation with papers based on other green technologies: high hydrostatic pressure, Pulsed electric field etc.

Answer: Thanks for your review and kind advice. We have added other green technologies (microwave, pulsed electric field, ultra-high pressure treatment) in the Introduction. Moreover, we have added relevant references in the manuscript. (Lines 59-64)

References:

  1. Cai, Luyun, Jianhui Feng, Ailing Cao, Yuhao Zhang, Yanfang Lv, and Jianrong Li. Denaturation Kinetics and Aggregation Mechanism of the Sarcoplasmic and Myofibril Proteins from Grass Carp During Microwave Processing. Food and Bioprocess Technology.2017, 11, 417-426.
  2. Wang, J. Y., Y. L. Yang, X. Z. Tang, W. X. Ni, and L. Zhou. Effects of Pulsed Ultrasound on Rheological and Structural Properties of Chicken Myofibrillar Protein. Ultrason Sonochem.2017, 38, 225-233.
  3. Liu, Qiaoyu, Zeqian Lin, Xiaomei Chen, Junwen Chen, Junshi Wu, Haiguang Chen, and Xiaofang Zeng. Characterization of Structures and Gel Properties of Ultra-High-Pressure Treated-Myofibrillar Protein Extracted from Mud Carp (Cirrhinus Molitorella) and Quality Characteristics of Heat-Induced Sausage Products. Lwt.2022, 165.

Q3. M&M overall. Due to the high amount of processes I suggest to add an illustrative scheme of Process layout for the MPs. This will make easier to understand in my opinion M&M section.

Answer: Thanks for your review and kind advice. We have added an illustrative scheme of process layout for the MPs. In addition, the illustrative scheme is listed in the M&M section.

Q4. Figure 1. The schematic diagram of the ultrasound device: I did not understand where is a US nozzle? Please indicate which part of your device was connected to US nozzle. Additionally please clarify which part of your chamber was connected to US nozzle (walls or bottom?).

Answer: Thanks for your review and kind advice. We have redrawn the schematic diagram of the ultrasound device. We have clarified that there was no US nozzle in our ultrasound equipment. (Figure.1)

Q5. Figure 2. Please add more discussion regarding U50 (T=54°C). Why does Surface hydrophobicity drops and this frequency level? Any oscillations?

Answer: Thanks for your review and kind advice. It has been added and is detailed in lines 198-201 in the manuscript.

We stated: The SUH was the least at the ultrasound frequency of 50 kHz when degree of doneness was Rare (52-55°C). In this case, excessive ultrasound frequency may cause cavitation, which re-embeds hydrophobic amino acids left on the protein surface. The conclusion of the study by Bian et al. [27] is similar.

Q6. Fig 3 and 5. There are several samples indicated as: UT1…UT6 and U20….U50 please clarify.

Answer: Thanks for your review and kind advice. We have added the relevant explanations about the treated samples.

Fig.3: U20: ultrasound frequency of 20 kHz; U25: ultrasound frequency of 25 kHz; U28: ultrasound frequency of 28 kHz; U40: ultrasound frequency of 40 kHz; U50: ultrasound frequency of 50 kHz.

Fig.5: (B) Control; (C) UT1 (ultrasound frequency of 20 kHz, 64℃); (D) UT2 (ultrasound frequency of 25 kHz, 64℃); (E) UT3 (ultrasound frequency of 28 kHz, 64℃); (F) UT4 (ultrasound frequency of  40 kHz, 64℃); (G) UT5 (ultrasound frequency of 50 kHz, 64℃); (H) UT6 (ultrasound frequency of 50 kHz, 72℃).

Q7. Line 416: The authors stated: When the ultrasonic frequency was 28 kHz, the surface hydrophobicity and sulfhydryl group contents of MPs under the same degree of doneness were the largest. However fig 2 do not state this information. Please clarify.

Answer: Thanks for your review and kind advice. We have not accurately described the results of the Fig. 2. We have rephrased the information of the Fig. 2.

We stated: When the ultrasonic frequency was 28 kHz, the surface hydrophobicity of MPs under the degree of doneness (Medium Well) were the largest. In addition, sulfhydryl group contents reached the largest value under r the degree of doneness (Rare, Medium Rare, Medium).

Reviewer 2 Report

In this paper, the authors investigate the impact of ultrasound frequency on the aggregation and structural changes of beef myofibrillar proteins with varying degrees of doneness. The results indicate that ultrasounds, in combination with the degree of doneness, affect the aggregation behavior and structural characteristics of beef myofibrillar proteins (MPs). The manuscript is well-written overall.

However, it is important for the authors to highlight the novelty of their research, as the use of ultrasounds for meat improvement is a well-known concept. The authors should emphasize the unique aspects or contributions of their study. While it is evident that ultrasounds and different degrees of doneness can alter the structural characteristics of beef myofibrillar proteins, the authors could explore other quality parameters of meat to provide a more comprehensive analysis.

Additionally, the quality of Figures 4 and 7 appears to be poor. The authors should consider improving the visual clarity and resolution of these figures to enhance their readability.

Finally, it is necessary for the authors to format the reference list according to the requirements specified by the Foods journal.

English language is OK.

Author Response

Response to reviewer #2’s comments

In this paper, the authors investigate the impact of ultrasound frequency on the aggregation and structural changes of beef myofibrillar proteins with varying degrees of doneness. The results indicate that ultrasounds, in combination with the degree of doneness, affect the aggregation behavior and structural characteristics of beef myofibrillar proteins (MPs). The manuscript is well-written overall.

However, it is important for the authors to highlight the novelty of their research, as the use of ultrasounds for meat improvement is a well-known concept. The authors should emphasize the unique aspects or contributions of their study. While it is evident that ultrasounds and different degrees of doneness can alter the structural characteristics of beef myofibrillar proteins, the authors could explore other quality parameters of meat to provide a more comprehensive analysis.

Additionally, the quality of Figures 4 and 7 appears to be poor. The authors should consider improving the visual clarity and resolution of these figures to enhance their readability.

Answer: Thanks for your review and kind advice. We have tried our best to improve the clarity of the Figures 4 and 7. We had revised in the Figures 4 and 7.

Finally, it is necessary for the authors to format the reference list according to the requirements specified by the Foods journal.

Answer: Thanks for your review and kind advice. We have checked all the references and revised the the writing style of the references according to the requirements specified by the Foods journal.

Round 2

Reviewer 1 Report

The authors answered my questions in current manner.

The manuscript now can be accepted.

Reviewer 2 Report

The authors corrected the manuscript accordingly.